# Proportion of School Attending Adolescents Meeting the Recommended Moderate-to-Vigorous Physical Activity Level and Its Predictors in Lagos, Nigeria

**DOI:** 10.3390/ijerph182010744

**Published:** 2021-10-13

**Authors:** Busola Adebusoye, Jo Leonardi-Bee, Revati Phalkey, Kaushik Chattopadhyay

**Affiliations:** Division of Epidemiology and Public Health, City Hospital Campus, University of Nottingham, Clinical Sciences Building, Nottingham NG5 1PB, UK; jo.leonardi-bee@nottingham.ac.uk (J.L.-B.); revati.phalkey@phe.gov.uk (R.P.); kaushik.chattopadhyay@nottingham.ac.uk (K.C.)

**Keywords:** physical activity, adolescents, schools

## Abstract

We aimed to assess the proportion of school attending adolescents who reached the recommended moderate-to-vigorous physical activity (MVPA) level in Lagos State, Nigeria, as well as the predictors associated with it. A cross-sectional study was conducted among 720 adolescents aged 12–19 years from 20 schools in Lagos State in 2020. MVPA level was assessed using the Activity Questionnaire for Adults and Adolescents. Predictors assessed were socio-demographic variables, anthropometric measurements, sedentary behaviour, self-efficacy, perceived benefits, and perceived barriers. Complete data was provided by 528 adolescents for the study (73% response rate). The recommended MVPA level was reached by 82.8% (95% CI 79.3–85.7) of the participants. Participants spent a median time of 44 (IQR 12.9, 110) minutes of MVPA per day on household-based activities, followed by school-based activities (21.4; 4.3, 50.4), active transportation (14.3; 0, 35), sport-based activities (8.6; 0, 58.9) and leisure-based activities (8.6; 1.1, 34.3). Participants in public schools were four times more likely to meet the recommended MVPA level compared to those in private schools (OR 3.97, 95% CI 2.46–6.42). A high proportion of school adolescents met the recommended MVPA level in Lagos State, Nigeria. Our study suggests that interventions for promoting MVPA should be targeted to private schools.

## 1. Introduction

The World Health Organization (WHO) recommends that adolescents should undertake at least 60 minutes of moderate-to-vigorous intensity physical activity (MVPA) per day [1]. MVPA helps adolescents to develop and maintain healthy musculoskeletal tissues, cardiovascular systems and body weight [1]. It improves their mental health by reducing depression, anxiety and stress [1]. It also improves their academic achievements and overall quality of life [2,3]. It assists in their social development by providing opportunities for self-expression, improving self-confidence, social interactions and integration [4]. Providing the opportunities for adolescents to engage in sufficient physical activity will prevent poor health outcomes such as increased adiposity, poorer fitness and reduced sleep duration [1].

In spite of the benefits of MVPA, data from 1.6 million students aged 11–17 years, which is equivalent to 81.3% of the global population of adolescents of this age, shows that only 19% of the world’s adolescents reach the recommended level [5]. This is even lower in Sub-Saharan Africa, where only 13.8% reached the recommended level in 2016 [5]. A recent study on adolescents meeting the recommended MVPA level reported data from only 16 out of 53 Sub-Saharan Africa countries, and no data were available on Nigerian adolescents [5]. Another study reported that 37% of adolescents in Nigeria reached the recommended MVPA level, however, this study was representative of only one state in the country [6] and did not assess the psychosocial correlates of physical activity among adolescents, such as self-efficacy, perceived benefits and perceived barriers that are known to be associated with physical activity among adolescents [7,8]. Furthermore, it did not consider whether the type of schools had an impact on the levels of physical activity among school attending adolescents. Schools represent a unique setting for the promotion of lifelong physical activity during critical development stages of life [9]. Opportunities for in-school physical activity are largely dependent upon school-level policies, practices and administrative support [10]. Existing evidence from systematic reviews demonstrates that school-based physical activity interventions account for significant improvement across several health outcomes for school going adolescents; and these interventions also increase students’ in school and out of school physical activity levels [11,12]. Very little is known on the predictors that could be important to formulate policies or design, evaluate and implement interventions to improve physical activity levels among Nigerian school attending adolescents [13]. Therefore, this study aimed to assess MVPA level among school attending adolescents in Lagos State and identify the predictors associated with it.

## 2. Materials and Methods

### 2.1. Study Design, Participants, Area and Period

A cross-sectional study was conducted among school attending adolescents aged 12–19 years in Lagos State, Nigeria. Students with learning disabilities (based on school records) and students aged < 12 years were excluded. Data were collected between February and March 2020.

### 2.2. Sampling and Sample Size

Lagos State has six districts of education, and each district is an agglomeration of three to four local governments in the state. District IV was selected because it is a mix of three local governments that are distinctly characterised into different socioeconomic strata [14,15]. Schools in this district were selected using stratified random sampling. In the first stage, schools were stratified by local government area (LGA) and then by school type using the master list accessed from the official internet portal of all schools in Lagos State [16]. Next, a random selection was made by probability proportional to their enrolment size by generating a random start number in Microsoft Excel. Subsequent schools were selected based on the sampling interval which was computed by dividing the total number of students in each LGA and schools by the number of schools needed. Selected schools were approached to participate in the study. If any school declined to participate, the next school on the list was approached. Using the proportion formula, we estimated that a sample size of 538 participants was adequate to detect a prevalence of 37% with a confidence level of 95%, 5% precision and 1.5 design effect [6,17]. Prior to data collection, the classes to be selected were assigned to each school to ensure representation of all age groups needed. All students in the class were invited to participate in the study. The number of students who declined to participate was recorded.

### 2.3. Data Collection Procedure and Tool

A self-reported quantitative questionnaire was developed and piloted among ten local students from a school who were not included in the final study. The questionnaire was administered to the participants in the class. It was available in English, the official language of the country. All participants received a physical activity themed exercise book as an appreciation of their time. The questionnaire contained three sections. Section I included questions on socio-demographic variables: date of birth (to calculate age in years), sex (male, female or prefer not to say), ethnicity (Hausa, Ibo, Yoruba or others), socioeconomic status (low, medium or high; assessed using the Material Affluence Scale (MAS) for low- and middle-income countries and contained questions on what property the family had, such as cars, fridge, television, computer, radio and home ownership, the number of people with whom the participants shared their rooms (dichotomised into ≤3 or more to indicate crowding) and the type of house they lived in (mud, bamboo, block, cemented and painted)) [18], type of school (public or private) and class (junior, first three years of the secondary school or senior, last three years of the secondary school).

Section II contained questions on physical activity and sedentary behaviour. MVPA level was assessed using the Activity Questionnaire for Adults and Adolescents (AQuAA) [19]. AQuAA shows acceptable evidence of test-retest reliability (ranged from 0.38 to 0.71) among adolescents in Nigeria [7]. MVPA level was computed by summing the time (minutes/week) spent on moderate and vigorous physical activities across the different domains. The proportion of participants that met the recommended MVPA level (i.e., 60 min of MVPA per day) was estimated. Sedentary behaviour was computed from the amount of time participants spent watching TV and surfing the internet per week. Those who had >14 h per week screen time (i.e., >2 h per day) were categorised as high sedentary behaviour [20].

Section III contained questions on self-efficacy, perceived benefits, and perceived barriers. Self-efficacy was assessed using a self-efficacy questionnaire [21]. The questionnaire asked participants to rate their agreement on their ability to be physically active in different situations on an eight-item scale. Each item used a 5-point Likert scale that ranged from strongly disagree to strongly agree. Perceived benefits and perceived barriers were assessed through questions used in similar studies [22,23,24]. Participants were asked to rate their agreement on a scale of strongly disagree to strongly agree on the effects (benefits) of physical activity (eight items, namely, physical appearance and weight, health and fitness, social interaction, pleasure, competition, relief from stress, admiration from others and relaxation). They were also asked to report the frequency of the barriers that prevented them from being physically active (13 items, namely, lack of time, lack of discipline, lack of interest, health problems, personal problems, not skilled enough, too expensive, no transportation, not liking to sweat, fear of being laughed at, cultural factors, climate not suitable and lack of facilities). Each response was graded on a one (strongly disagree) to five (strongly agree) scale. Mean scores were computed for each participant for each of these scales [25].

Anthropometric measurements were recorded by the researchers. Participants’ body weight and height were measured twice using the DETECTO slimPRO (Cardinal/DETECTO, MO, USA) and Seca 213 (Seca, Hamburg, Germany) to the nearest 0.1 kg and 0.1 cm, respectively. Body Mass Index (BMI) was computed as mean body weight (kg) divided by mean height (m) squared (kg/m^2^). Age and sex specific prevalence of grade I-III thinness, normal, overweight and obesity was determined using the WHO criteria and with the zanthro package in STATA V.14.2 (Stata Corp LLC, College Station, TX, USA) [26]. Participants aged 18 to 19 years were classified according to the adult WHO BMI classification [27]. Waist and hip circumference were measured twice with Seca 203 (Seca, Hamburg, Germany) to the nearest 0.1 cm. The waist-to-hip ratio was computed by dividing the mean waist circumference by the mean hip circumference.

### 2.4. Data Analysis

Frequencies and proportions were reported for categorical variables. Means and standard deviations were reported for normally distributed continuous variables, and medians and interquartile range for non-normally distributed continuous variables. To assess the representativeness of the respondents included in the analyses, the characteristics of participants with and without missing outcome data (MVPA level) were compared using the chi-squared test for categorical variables and *t*-tests for continuous variables. Where predictors could be added to the model as either a continuous or categorical variable, we fitted the predictor as a continuous variable where there was a significant linear trend across categories. To deal with missing predictor values, we included a separate category for missing data for each categorical predictor and for each continuous predictor, we assigned the mean value of each continuous variable to the missing value and included a dummy variable in the model indicating the presence or absence of missing data. Univariate logistic regression was conducted to investigate the crude association between the proportion of participants that met the recommended MVPA level and predictor variables. To identify any independent association, multivariable logistic regression was performed, where predictors with a *p*-value ≤ 0.10 in the univariate logistic regression were initially included. Next, all non-statistically significant predictors in the model were removed and added independently to a model that consisted of only the significant predictors. The final model consisted of the predictor variables that were statistically significant, (*p* ≤ 0.05). The crude and adjusted odds ratios (ORs) together with the 95% confidence intervals (CIs) and *p*-values are presented. Data were analysed using STATA V.14.2 (Stata Corp LLC, College Station, TX, USA) [28].

## 3. Results

A total of 33 schools (10 public and 23 private) were approached to participate in the study, and 11 private schools declined the request. Two private schools that initially expressed an interest to participate did not take part due to the COVID-19 pandemic. The remaining 20 schools (10 public and 10 private) participated. A total of 752 students were sampled. Figure 1 shows the flowchart of the study participants. Participants without outcome data (MVPA level) tended to be older (*p* = 0.005), were more likely to have a higher socioeconomic status (*p* = 0.025) and be in the senior class (*p* < 0.001). (Appendix A).

Participants’ characteristics are reported in Table 1. The mean age of the participants was 14.9 ± 1.6 years, and 55.3% of them were female. Of the participants, 82.8% (95% CI 79.3 to 85.7) met the recommended MVPA level. Participants spent a median time of 44 (IQR 12.9, 110) minutes of MVPA per day on household-based activities, followed by school-based activities (21.4; 4.3, 50.4), active transportation (14.3; 0, 35), sport-based activities (8.6; 0, 58.9) and leisure-based activities (8.6; 1.1, 34.3).

Table 2 reports the unadjusted association between meeting the recommended MVPA level and predictors. Participants who were older, had a higher BMI, had a higher waist-to-hip ratio, had a lower socioeconomic status, attended a public school or had higher self-efficacy were significantly more likely to meet the recommended MVPA level.

Table 3 reports the association between meeting the recommended MVPA level and predictors. Following the model building strategy, the only significant predictor was school type, where participants who attended a public school were 3.97 times more likely to meet the recommended MVPA level compared to those in private schools (OR 3.97, 95% CI 2.46 to 6.42).

## 4. Discussion

This study presents the proportion of school attending adolescents who reached the WHO recommended MVPA level in Lagos State, Nigeria. A total of 82.8% of our participants reached the recommended MVPA level, and this was higher than what was reported in the northern part of the country (i.e., 37%) [6]. This wide disparity could be due to the ownership of cars as more than 73% of their participants reported household ownership of one or more cars compared to 54% in our study. This study additionally reported that adolescents with family ownership of cars reported significantly less active transportation. We found certain similarities between the two studies—boys participated in sports more than girls, while girls were more active than boys in home-based activities [6]. This finding of higher participation in sports among boys compared to girls is similar to what was reported in a study in South Africa, where boys spent more time on vigorous sports than girls [29], and to a study that compared physical activity patterns between boys and girls across seven countries in Sub-Saharan Africa [30]. The proportion of school attending adolescents who reached the recommended MVPA level in our study is higher than what has been reported globally [5]. We used a questionnaire that assessed physical activity across all the domains and the time spent on each, and this could have led to higher MVPA level reporting. Other studies assessed physical activity based on participation in sports by asking the question, ‘over the last seven days, how many days were you physically active for a combined total of at least 60 min per day’ [30,31,32]. For direct comparison, 35.7% of our study participants reported meeting the recommended MVPA level through sports.

Although we assessed the independent effect of various predictors on MVPA level, we found the type of school was the only significant predictor, where adolescents in public schools were more likely to reach the recommended MVPA level compared to those in private schools. This result is similar to what was reported among school-going children in Kampala, Uganda [33] and school-going adolescents in Jordan [34]. Similarly, in a study conducted in Saudi Arabia, girls in public secondary schools were more physically active than girls in private schools [35]. A plausible reason for this could be socioeconomic differences [35]. In our study, students in private schools were of higher socioeconomic status than those in public schools (78% vs. 38%). Hence, the chances of private school students doing household activities or using active transportation were less, and this could have contributed to their lower MVPA level. Furthermore, being in a private school could be an indication of having parents who place more emphasis on education and higher academic achievements than on sports and active recreation [5]. This has often been cited as a barrier to participation in physical activities among adolescents [5]. Also, having visited the schools for study data collection, public schools had a larger expanse of land compared to private schools.

Finally, our finding in relation to public school students being at increased odds of reaching the recommended level compared to private school students’ conflict with findings from high income countries where private school students are more likely to have access to more physical activity opportunities than public school students due to access to more financial resources [36]. However, this is not the case in Lagos State, many of the private schools in Lagos State, just like in many other low- and middle-income countries (LMICs), are created to serve low-income families because the public education systems often face substantial constraints [37]. Parents choose private schools because government schools are perceived to be failing or too far from home. Such private schools run on incredibly tight budgets with often untrained teachers and therefore cannot provide the adequate resources for physical activity [38].

### Strengths and Limitations

To the best of our knowledge, this was the first study to assess MVPA level among school attending adolescents in Lagos, Nigeria. We included participants from both high- and low-income groups. This, therefore, makes the findings generalisable to the school attending adolescents in Lagos State. Similar studies should be conducted in other parts of the country to have a complete picture of the issue. Our study suggests the physical activity domains where physical activity interventions should be targeted.

The study is, however, not without its limitations. The assessment of MVPA level was based on self-reporting, and this could have introduced social desirability bias and over-estimated MVPA level of participants. Although the questionnaire used to assess MVPA level showed an acceptable test-retest reliability among adolescents in Nigeria [7], there was a lot of missing data in our study. This is probably due to the nature of the questionnaire which required three pieces of information (the number of days on which the activity was performed, the duration, and intensity with which the activity was performed) to determine an answer. The failure to provide one out of the three required pieces of information led to missing data. Therefore, future studies should consider using objective measurement tools to assess MVPA level and validated questionnaires requiring less information to assess MVPA level, if self-reported.

## 5. Conclusions

In conclusion, there is a high proportion of school attending adolescents meeting the recommended MVPA level in Lagos, Nigeria. The lower MVPA level seen in private schools calls for a synergistic approach from all the stakeholders such as the government, school administrators, researchers, and parents to seek ways of promoting the importance of MVPA as well as the avenues for students to engage in it.

## Figures and Tables

**Figure 1 ijerph-18-10744-f001:**
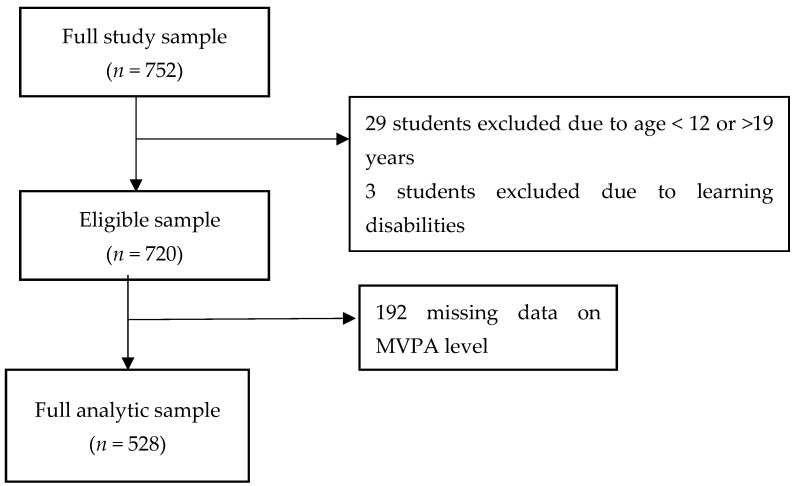
Flowchart of the study participants.

**Table 1 ijerph-18-10744-t001:** Characteristics of participants.

		Meeting the Recommended MVPA Level	
Characteristics	Total *n* = 528 *n* (%) or Otherwise Indicated	Yes *n* = 437 *n* (%) or Otherwise Indicated	No *n* = 91 *n* (%) or Otherwise Indicated
Age (years) ^a^	14.9 (1.6) ^b^	14.9 (1.6) ^b^	14.5 (1.5) ^b^
Sex			
Male	234 (44.3)	194 (44.4)	40 (44.0)
Female	292 (55.3)	241 (55.2)	51 (56.0)
Prefer not to say	2 (0.4)	2 (0.5)	0 (0)
Ethnicity			
Hausa	6 (1.1)	5 (1.1)	1 (1.1)
Ibo	172 (32.6)	133 (30.4)	39 (42.9)
Yoruba	291 (55.1)	247 (56.5)	44 (48.4)
Others	58 (11.0)	51 (11.7)	7 (7.7)
Missing	1 (0.2)	1 (0.2)	0 (0)
Socioeconomic status			
Low	229 (43.4)	204 (46.7)	25 (27.5)
Middle	213 (40.3)	167 (38.2)	46 (50.6)
High	75 (14.2)	57 (13.0)	18 (19.8)
Missing	11 (2.1)	9 (2.1)	2 (2.2)
School			
Public	319 (60.4)	289 (66.1)	30 (33.0)
Private	209 (39.6)	148 (33.9)	61 (67.0)
Class			
Junior	231 (43.8)	197 (45.1)	34 (37.4)
Senior	297 (56.3)	240 (54.9)	57 (62.6)
BMI			
Grade I-III thinness	111 (21.0)	98 (22.4)	13 (14.3)
Normal	356 (67.4)	296 (67.7)	60 (65.9)
Overweight	51 (9.7)	37 (8.5)	14 (15.4)
Obese	8 (1.5)	5 (1.1)	3 (3.3)
Missing	2 (0.4)	1 (0.2)	1 (1.1)
Waist-to-hip ratio	0.8 (0.04) ^b^	0.80 (0.04) ^b^	0.80 (0.05) ^b^
Missing	19		
Sedentary behaviour			
Low	214 (40.5)	179 (41.0)	35 (38.5)
High	290 (54.9)	237 (54.2)	53 (58.2)
Missing	24 (4.6)	21 (4.8)	3 (3.3)
Self-efficacy	3.5 (0.6) ^b^	3.6 (0.6) ^b^	3.4 (0.6) ^b^
Missing	2		
Perceived benefits	3.8 (0.6) ^b^	3.8 (0.6) ^b^	3.7 (0.6) ^b^
Missing	2		
Perceived barriers	2.6 (0.6) ^b^	2.6 (0.6) ^b^	2.7 (0.7) ^b^
Missing	2		
**Time Spent on MVPA across the Different Domains**			
Household	44 (12.9, 110) ^c^	60 (25, 128.57) ^c^	7.1 (2, 15) ^c^
School	21.4 (4.3, 50.4) ^c^	25 (7.14, 64.29) ^c^	4.3 (0, 10.7) ^c^
Active transportation	14.3 (0, 35) ^c^	17.9 (0.7, 42.8) ^c^	1.4 (0,10.7) ^c^
Sport	8.6 (0, 58.9) ^c^	17.1 (0, 70) ^c^	0 (0, 4.57) ^c^
Leisure	8.6 (1.1, 34.3) ^c^	13.6 (2.9, 45) ^c^	1 (0, 5) ^c^

^a^ Age was included as a continuous predictor; BMI, Body Mass Index; MVPA, Moderate-to-vigorous physical activity; ^b^ Mean (SD); ^c^ Median (IQR).

**Table 2 ijerph-18-10744-t002:** Unadjusted association between meeting the recommended MVPA level and predictors.

Characteristics	Unadjusted OR	95% CI	*p*-Value
Age (years)	1.21	1.04–1.42	0.016
Sex			
Male	1.00		
Female	0.97	0.62–1.54	0.911
Prefer not to say	Not estimable		
Missing	Not estimable		
Ethnicity			0.142
Yoruba	1.00		
Hausa	0.89	0.10–7.81	
Ibo	0.61	0.38–0.98	
Others	1.30	0.55–3.04	
Missing	Not estimable		
Socioeconomic status			0.007
Low	1.00		
Middle	0.45	0.26–0.76	
High	0.39	0.20–0.76	
Missing	0.55	0.11–2.70	
School			<0.001
Private	1.00		
Public	3.97	2.46–6.42	
Class			0.175
Junior	1.00		
Senior	0.73	0.46–1.16	
BMI			0.063
Normal	1.00		
Grade I-III thinness	1.53	0.80–2.90	
Overweight	0.54	0.27–1.05	
Obese	0.34	0.08–1.45	
Missing	0.20	0.01–3.29	
Waist-to-hip ratio	0.01	0.00–2.07	0.091
Missing	Not estimable		
Sedentary behaviour			0.689
Low	1.00		
High	0.87	0.55–1.40	
Missing	1.37	0.39–4.84	
Self-efficacy	1.44	1.02–2.03	0.039
Missing	Not estimable		
Perceived benefits	1.28	0.90–1.81	0.171
Missing	Not estimable		
Perceived barriers	0.90	0.63–1.30	0.583
Missing	0.62	0.06–6.05	0.683

BMI, Body Mass Index.

**Table 3 ijerph-18-10744-t003:** Independent association between meeting the recommended MVPA level and predictor.

Characteristics	OR	95% CI	*p*-Value
School			<0.001
Private	1.00		
Public	3.97	2.46–6.42	

## Data Availability

A de-identified data set will be available upon request unless there are legal or ethical reasons for not doing so.

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
