# Peer review of "Proportion of School Attending Adolescents Meeting the Recommended Moderate-to-Vigorous Physical Activity Level and Its Predictors in Lagos, Nigeria"

_ijerph, 2021, doi:10.3390/ijerph182010744_

Round 1

Reviewer 1 Report

The introduction is too short, the background of the problem needs to be elaborated more. 

In chapter materijals and methods there are 4 subchapter but there are no subchapter sample. It can be very short with the sentence detailed data on the sample can be found in Table 1 

The type of school (private public) is significantly related to the MVPA and the socioeconomic status is not. In most countries, private schools are exclusively for students from high-income group. Is that the case in Lagos as well, that needs comment. 

Author Response

Dear Editor and Reviewers,

Thank you very much for considering our manuscript. We have amended the manuscript based on the comments. We have highlighted the changes (in yellow) in the manuscript.

We look forward to hearing from you.

Best regards,

Busola Adebusoye

Reviewer 1:

  1. The introduction is too short, the background of the problem needs to be elaborated more.

Response: We thank the reviewer for this comment. We have now elaborated the background of the problem through including additional information to the introduction.

From line 29-36, we have added information on the consequences of physical inactivity and the benefits associated with physical activity.

MVPA helps adolescents to develop and maintain healthy musculoskeletal tissues, cardiovascular system and body weight [1]. It improves their mental health by reducing depression, anxiety and stress [1]. It also improves their academic achievements and overall quality of life [2,3]. It assists in their social development by providing opportunities for self-expression, improving self-confidence, social interactions and integration [4]. Providing the opportunities for adolescents to engage in sufficient physical activity will prevent poor health outcomes such as increased adiposity, poorer fitness and reduced sleep duration [1].

From line 37-41, we have described the global levels of MVPA in adolescents.

Data from 1.6 million students aged 11-17 years, which is equivalent to 81.3% of the global population of adolescents of this age, shows that only 19% of the world’s adolescents reach the recommended level [5]. This is even lower in Sub-Saharan Africa as only 13.8% of them reached the recommended level in 2016 [5].

From line 49-56, we have elaborated more on schools as settings to promote physical activity.

Schools represent a unique setting for the promotion of lifelong physical activity during critical development stages of life [9]. Opportunities for in-school physical activity are largely dependent upon school-level policies, practices and administrative support [10]. Existing evidence from systematic reviews demonstrates that school-based physical activity interventions account for significant improvement across several health outcomes for school going adolescents; and these interventions also increase students’ in school and out of school physical activity levels [11,12].

  1. In chapter materials and methods there are 4 subchapters but there are no subchapter sample. It can be very short with the sentence detailed data on the sample can be found in Table 1

Response: Thank you for this point; however, the second subchapter is entitled ‘Sampling and sample size’, which includes information on the sample and sample size estimation.

From line 67-84:

2.2. Sampling and sample size

Lagos State has six districts of education, and each district is an agglomeration of three to four local governments in the state. District IV was selected because it is a mix of three local governments that are distinctly characterised into different socioeconomic strata [14,15]. Schools in this district were selected using stratified random sampling. In the first stage, schools were stratified by local government area (LGA) and then by school type using the master list accessed from the official internet portal of all schools in Lagos State [16]. Next, a random selection was done by probability proportional to their enrolment size by generating a random start number in Microsoft Excel. Subsequent schools were selected based on the sampling interval which was computed by dividing the total number of students in each LGA and schools by the number of schools needed. Selected schools were approached to participate in the study. If any school declined to participate, the next school on the list was approached. Using the proportion formula, we estimated that a sample size of 538 participants was adequate to detect a prevalence of 37% with a confidence level of 95%, 5% precision and 1.5 design effect [6,17]. Prior to data collection, the classes to be selected were assigned to each school to ensure representation of all age groups needed. All students in the class were invited to participate in the study. The number of students who declined to participate was recorded.

Additionally, we have provided detailed information regarding the sample from line 168-173.

Participants’ characteristics are reported in Table 1. The mean age of the participants was 14.9±1.6 years, and 55.3% of them were female. 82.8% (95% CI 79.3 to 85.7) of the participants met the recommended MVPA level. Participants spent a median time of 44 (IQR 12.9, 110) minutes of MVPA per day on household-based activities, followed by school-based activities (21.4; 4.3, 50.4), active transportation (14.3; 0, 35), sport-based activities (8.6; 0, 58.9) and leisure-based activities (8.6; 1.1, 34.3).

  1. The type of school (private public) is significantly related to the MVPA and the socioeconomic status is not. In most countries, private schools are exclusively for students fromhigh-income group. Is that the case in Lagos as well, that needs comment. 

Response: Thank you for this important point. In the discussion section, we have described the peculiarities of private schools in Lagos State, explaining that many private schools are make-shifts and not for the ultra-rich like it is in high income countries. From line 228-238

Finally, our finding in relation to public school students being at increased odds of reaching the recommended level compared to private school students conflict with findings from high income countries where private school students are more likely to have access to more physical activity opportunities than public school students due to access to more financial resources [36]. However, this is not the case in Lagos State, many of the private schools in Lagos State just like in many other low- and middle-income countries (LMICs), are created to serve low-income families because the public education systems often face substantial constraints [37]. Parents choose private schools because government schools are perceived to be failing or too far from home. Such private schools run on incredibly tight budgets with often untrained teachers and therefore cannot provide the adequate resources for physical activity [38].

Reviewer 2 Report

The Authors decided to investigate the proportion of school attending adolescents who reached the recommended amount of moderate-to-vigorous physical activity. The problem of physical activity, especially among children and adolescents is a very important and interesting issue. Another strength of the study, that should be pointed out is the significant number of individuals included in the examined group.

Detailed comments regarding each part of the text:

  • Introduction: Articles cited in the introduction part of the study consider mainly the Nigerian population, which is justified considering the study group. However, in the Reviewer’s opinion, it will be helpful to also present basic information regarding the problem of physical activity among youth in other populations.
  • Materials and methods:
    • thinness is described as stage III to I, which is justified considering the fact that stage III represents the most severe thinness; however, in the Reviewer’s opinion, the degrees of thinness should be presented as I to III;
    • only the first word is capitalised in the full name of BMI, while all words of this name should be capitalised - Body Mass Index;
    • line 53 – it is stated, that “Nigeria. Students with learning disabilities (based on school records)were excluded” – were there any other inclusion or exclusion criteria? For example, based on physical disability?
  • Results: In the table presenting the characteristics of the study group the number of males and females in the study group is presented under “gender”. Though, considering that the study group was, most probably, divided based on morphological and not psychological characteristics, the term “sex” instead of gender should be used.
  • Discussion: In the Reviewer’s opinion it will be helpful to present some additional pieces of information regarding the consequences of insufficient activity as well as benefits associated with physical activity.

Author Response

Dear Editor and Reviewers,

Thank you very much for considering our manuscript. We have amended the manuscript based on the comments. We have highlighted the changes (in yellow) in the manuscript.

We look forward to hearing from you.

Best regards,

Busola Adebusoye

The Authors decided to investigate the proportion of school attending adolescents who reached the recommended amount of moderate-to-vigorous physical activity. The problem of physical activity, especially among children and adolescents is a very important and interesting issue. Another strength of the study, that should be pointed out is the significant number of individuals included in the examined group.

Detailed comments regarding each part of the text:

  1. Introduction: Articles cited in the introduction part of the study consider mainly the Nigerian population, which is justified considering the study group. However, in the Reviewer’s opinion, it will be helpful to also present basic information regarding the problem of physical activity among youth in other populations.

Response: Thank you for this important comment. We have now included additional information to the introduction. From line 37-41, we have described the global levels of MVPA in adolescents.

Data from 1.6 million students aged 11-17 years, which is equivalent to 81.3% of the global population of adolescents of this age, shows that only 19% of the world’s adolescents reach the recommended level [5]. This is even lower in Sub-Saharan Africa as only 13.8% of them reached the recommended level in 2016 [5].

Additionally, from line 49-56, we have elaborated more generally on school as settings to promote physical activity.

Schools represent a unique setting for the promotion of lifelong physical activity during critical development stages of life [9]. Opportunities for in-school physical activity are largely dependent upon school-level policies, practices and administrative support [10]. Existing evidence from systematic reviews demonstrates that school-based physical activity interventions account for significant improvement across several health outcomes for school going adolescents; and these interventions also increase students’ in school and out of school physical activity levels [11,12].

  1. Materials and methods:
    1. thinness is described as stage III to I, which is justified considering the fact that stage III represents the most severe thinness; however, in the Reviewer’s opinion, the degrees of thinness should be presented as I to III;

Response: Thank you for highlighting this error. This has now been corrected and highlighted in all the places where they appear: Line 130, Table 1, Table 2, and Table S1.

    1. only the first word is capitalised in the full name of BMI, while all words of this name should be capitalised - Body Mass Index

Response: We apologise for this error. This has now been corrected in all the places where they appear line 128, line 176, line 183.

    1. line 53 – it is stated, that “Nigeria. Students with learning disabilities (based on school records) were excluded” – were there any other inclusion or exclusion criteria? For example, based on physical disability?

Response: Thank you for this comment. We have expanded the exclusion criteria to clarify that students less than 12 years old were also excluded. We did not exclude anyone based on physical disability.

Line 65.

Students with learning disabilities (based on school records) and students aged < 12 years were excluded.

  1. Results: In the table presenting the characteristics of the study group the number of males and females in the study group is presented under “gender”. Though, considering that the study group was, most probably, divided based on morphological and not psychological characteristics, the term “sex” instead of gender should be used.

Response: We apologise for this error. The term “gender” has been replaced with “sex” in all the places where they appear: line 92, Table 1, Table 2 and Table S1.

  1. Discussion: In the Reviewer’s opinion it will be helpful to present some additional pieces of information regarding the consequences of insufficient activity as well as benefits associated with physical activity.

Response: Thank you for this important point. Additional information regarding the consequences of physical inactivity and the benefits associated with physical activity have now been provided in the introduction from line 29-36.

MVPA helps adolescents to develop and maintain healthy musculoskeletal tissues, cardiovascular system and body weight [1]. It improves their mental health by reducing depression, anxiety and stress [1]. It also improves their academic achievements and overall quality of life [2,3]. It assists in their social development by providing opportunities for self-expression, improving self-confidence, social interactions and integration [4]. Providing the opportunities for adolescents to engage in sufficient physical activity will prevent poor health outcomes such as increased adiposity, poorer fitness and reduced sleep duration [1].